# Effects of Through-Bond and Through-Space Conjugations on the Photoluminescence of Small Aromatic and Aliphatic Aldimines

**DOI:** 10.3390/molecules27228046

**Published:** 2022-11-19

**Authors:** Peifeng Zhuang, Chang Yuan, Yunhao Bai, Changcheng He, Jiayu Long, Hongwei Tan, Huiliang Wang

**Affiliations:** 1Beijing Key Laboratory of Energy Conversion and Storage Materials, College of Chemistry, Beijing Normal University, Beijing 100875, China; 2Key Laboratory of Theoretical and Computational Photochemistry, Ministry of Education, College of Chemistry, Beijing Normal University, Beijing 100875, China

**Keywords:** through-space conjugation, through-bond conjugation, aldimines, nontraditional luminogens

## Abstract

Through-bond conjugation (TBC) and/or through-space conjugation (TSC) determine the photophysical properties of organic luminescent compounds. No systematic studies have been carried out to understand the transition from aromatic TBC to non-aromatic TSC on the photoluminescence of organic luminescent compounds. In this work, a series of small aromatic and aliphatic aldimines were synthesized. For the aromatic imines, surprisingly, *N*,1-diphenylmethanimine with the highest TBC is non-emissive, while *N*-benzyl-1-phenylmethanimine and *N*-cyclohexyl-1-phenylmethanimine emit bright fluorescence in aggregate states. The aliphatic imines are all emissive, and their maximum emission wavelength decreases while the quantum yield increases with a decrease in steric hindrance. The imines show concentration-dependent and excitation-dependent emissions. Theoretical calculations show that the TBC extents in the aromatic imines are not strong enough to induce photoluminescence in a single molecule state, while the intermolecular TSC becomes dominant for the fluorescence emissions of both aromatic and aliphatic imines in aggregate states, and the configurations and spatial conformations of the molecules in aggregate states play a key role in the formation of effective TSC. This study provides an understanding of how chemical and spatial structures affect the formation of TBC and TSC and their functions on the photoluminescence of organic luminescent materials.

## 1. Introduction

Traditional organic photoluminescent materials generally contain conventional chromophores such as large π-conjugated benzene rings and/or heterocycles. The early luminogens generally contain planar π-conjugated systems and exhibit aggregation-caused quenching (ACQ) behavior in concentrated solutions and solid state, which strongly impedes their practical applications. In 2001, Tang’s group [1] discovered and developed a novel concept of aggregation-induced emission (AIE), i.e., the luminogens are weakly or even non-emissive in dilute solutions (discrete state) but strongly emissive in the aggregate states. Typically, AIEgens such as tetraphenylethene (TPE) and hexaphenylsilole (HPS) contain nonplanar π-conjugated systems. The restriction of intramolecular motion (RIM) in the aggregate state is employed to explain the AIE behaviors of AIEgens [2,3,4,5]. In recent years, a large number of natural and synthetic polymers and small organic compounds without any conventional chromophores have been reported to exhibit intrinsic photoluminescence [6,7,8,9,10,11,12,13,14,15,16,17,18,19,20,21]. These nontraditional luminogens (NTLs) generally contain electron-rich heteroatoms such as N, O, S, and B and/or unsaturated bonds such as C=C, C=O and C≡N. These NTLs possess many impressive advantages, including easy preparation, structural stability, hydrophilicity, and biocompatibility, which make them ideal candidates for sensors and biological and medical applications.

In traditional organic luminogens, conventional chromophores are large π-conjugated structures formed by through-bond conjugation (TBC). TBC plays a dominant role in the photoluminescence of traditional luminogens. The increase in TBC extent reduces the HOMO-LUMO energy gap of the molecule so that it is easy to be excited, and the excited state energy is easily released in the form of radiation [22]. Researchers have found that intra/intermolecular through-space conjugation (TSC) also has an important impact on the fluorescence emission of traditional luminogens [23,24]. Zhang et al. [25] reported the AIE behaviors of 1,1,2,2-tetraphenylethane (*s*-TPE) and 1,1,2,2-tetrakis(2,4,5-trimethyl phenyl)ethane (*s*-TPE-TM). They found that the emissions of these two compounds are red-shifted in the aggregate state with comparison to in the discrete state, and that the *s*-TPE has a higher quantum yield and more red-shifted emission than *s*-TPE-TM. Theoretical calculations show that the overlap of phenyl groups in the aggregates of *s-*TPE and *s*-TPE-TM leads to a significant TSC effect, which induces the lowered HOMO-LUMO gap and hence the red-shifted emission. Comparatively, due to the steric hindrance of methyl groups on *s*-TPE-TM, its intramolecular TSC is weaker than that of *s*-TPE. Introducing electron-donating groups into triphenylmethane (TPM) increases electronic density and stabilizes TSC, leading to red-shifted emission [26].

For ACQ compounds, they have very large TBC due to their rigid planar π-conjugated systems; therefore, they can emit strong fluorescence in a single molecule state. However, in the aggregate state, strong TSC can also be formed between molecules, and the combination of strong TBC and TSC leads to the formation of excimer/exciplex of excited molecules, which leads to fluorescence quenching. On the contrary, if the TBC extent of the compound itself is not high or the compound does not even have TBC, the combination of TBC and TSC or only TSC can lead to fluorescence emission. In addition, both TSC and TBC can rigidify the molecular conformation and hence inhibit the non-radiative transition of molecules, which is beneficial to fluorescence emissions.

The photoluminescence of NTLs is explained with clusteroluminescence (CL) [27] and Clustering-Triggered Emission (CTE) [8,28] mechanisms. Due to the lack of large TBC, especially aromatic TBC, the intra/intermolecular TSC through n–n, n–π, and/or π−π interactions in clusters of nonconventional chromophores becomes the main interaction for the photoluminescence of NTLs. Many theoretical studies and experimental evidence have proven the existence of TSC and its function in the photoluminescence of NTLs.

Organic luminescent compounds are commonly categorized into traditional and nontraditional luminescent compounds based on the presence or absence of large π-conjugated moieties (e.g., aromatic rings). However, there should be no strict boundary between these two types, because conventional and nonconventional chromophores can exist simultaneously in many organic luminescent compounds. The key problem involved in the transition from traditional luminescent compounds to NTLs is the transition of the types of TBC and/or TSC conjugations. Up to now, no systematic studies have been carried out to understand the transition from aromatic TBC to non-aromatic TSC on the photoluminescence of organic luminescent compounds.

Some aromatic imines (Schiff bases) have already been reported to exhibit AIE behavior [29,30,31]. In recent years, a few aliphatic aldimines [32] and ketimines [33] are also reported to exhibit intrinsic fluorescence. In this work, a series of small aromatic and aliphatic aldimines with different phenyl and alkyl substituents connected to the C=N bond were synthesized. Due to the presence or absence of aromatic TBC as well as the different extents of intermolecular TSC induced by the configurations and conformations of these compounds, the photophysical properties of these aldimines vary from sample to sample. The photophysical properties of these aldimines were systematically studied and were correlated with the chemical structures and intermolecular interactions using theoretical calculations.

## 2. Results and Discussion

### 2.1. Synthesis and Characterization of Aldimines

Through the condensation reactions between different primary amines and aldehydes, six aldimines were successfully synthesized (Figure 1). They are numbered as imines **1**–**6**. For the aromatic imines **1**–**3**, one phenyl group is always attached to the C atom of the C=N group, while the group attached to the N atom of the C=N group is phenyl, benzyl, and cyclohexyl, respectively. Due to the insertion of a methylene group between a phenyl group and the C=N group in imine **2** and the replacement of a phenyl group with an aliphatic cyclohexyl group in imine **3**, their TBC is possibly lower than that of imine **1**. Moreover, for the aliphatic imines, the alkyl groups attached to the C=N group are cyclohexyl, *n*-butyl, and isobutyl. Cyclohexyl group has a bigger steric hindrance and a stronger electron-donating ability than the *n*-butyl and isobutyl groups.

The synthesis of aldimines was proved using chemical characterizations with ^1^H NMR, FT-IR, and high-resolution MS (Appendix A). The ^1^H NMR chemical shifts of the hydrogen atom in the -N=CH group of the imines **1**–**6** are 8.47, 8.41, 8.31, 7.44, 7.38, and 7.40 ppm, respectively. Moreover, the FT-IR absorption bands of the C=N group in the imines are at 1625, 1644, 1645, 1665, 1668, and 1671 cm^−1^, respectively. These characterizations also prove that the aldimines exist in imine rather than enamine form. Note that the chemical shift of -N=CH decreases while the wavenumber of the C=N group increases with the decrease in the number of phenyl group for the aromatic imines, and there is a big decrease in the chemical shift in the -N=CH group while a big increase in the wavenumber of the C=N group in the Fourier transform infrared (FTIR) spectra for the aliphatic imines with comparison to the aromatic imines. A slight increase in the wavenumber of the C=N group with the decrease in steric hindrance is also observed for the aliphatic imines.

UV-visible (UV-vis) spectra of the imines **1**–**6** are shown in Figure 1a. For the imine **1** with a larger TBC system, the π→π* transition (260 nm) of the conjugated structure of phenyl is red-shifted in comparison to the π→π* transition (245 nm) in the imines **2** and **3**. The absorption band of imine **1** at 300–350 nm should be attributed to the π→π* electronic transition of conjugated phenyl and the C=N groups. For the aliphatic imines **4–6**, the absorption band at about 230 nm is attributed to the n→π* transition of the C=N group. Figure 1b shows the UV-vis spectra of imine **4** ethanol solutions with different concentrations, and a wide new absorption band appears at wavelengths larger than 300 nm, with two shoulder peaks at about 385 nm and 410 nm; moreover, its absorbance gradually increases with increasing concentration. Similar results are also observed for other imines (Appendix A). The appearance of new absorption bands at longer wavelengths at high concentrations suggests the aggregation of imine molecules and the formation of the clusters of the C=N groups and hence the formation of intermolecular TSC, which results in the red-shifted UV absorption wavelength and increased absorbance.

By comparing the UV absorption spectra of the aldimines, it can be found that, at a low wavelength (<350 nm), at the same wavelength, the absorbance of the solutions of aromatic aldimines, especially imine **3**, is much higher than that of the aliphatic aldimines at the same concentration. However, the absorbances of aliphatic imines **4**–**6** at around 400 nm are similar to or even higher than those of the aromatic imines **2** and **3**.

### 2.2. Photoluminescence Behaviors of Aldimines

The photophysical properties of the imines **1**–**6** were studied. Among the six imines, only imine **1** is solid at room temperature, and the other five are liquid. It is surprising to find that, at room temperature, imine **1** is non-emissive either in solid state or in dilute solutions, while the imines **2**–**6** show strong fluorescence in pure liquid state and in solutions (Figure 2a,b). The lifetimes of the emissions are in nanoseconds, indicating the fluorescence nature of the emissions (Appendix A). The emission colors of the ethanol solutions of imines **2**–**6** (5 vol.%) under 365 nm UV irradiation vary from sample to sample, and they appear in a dark-blue to light-green region (Figure 2a). The fluorescence emission spectra of pure imines **2**–**6** in a pure liquid state and their maximum excitation wavelength ($\lambda_{\mathrm{ex}}^{\max}$) and maximum emission wavelength ($\lambda_{\mathrm{em}}^{\max}$) are shown in Figure 2c,d, respectively. Imine **3** shows the highest $\lambda_{\mathrm{ex}}^{\max}$ and $\lambda_{\mathrm{em}}^{\max}$, while those of imine **6** are the lowest.

The fluorescence excitation and emission spectra of the ethanol solutions of imines **2**–**6** with different concentrations were also measured. The imines show an intriguing concentration-dependent emission (CDE). Generally, with an increase in concentration, the photoluminescence (PL) intensity increases, and the $\lambda_{\mathrm{ex}}^{\max}$ and $\lambda_{\mathrm{em}}^{\max}$ also red-shift (Figure 3). It is worth noting that, for the imines **2**, **3,** and **5**, their $\lambda_{\mathrm{em}}^{\max}$ keep almost constant firstly; however, an abrupt red-shift of $\lambda_{\mathrm{em}}^{\max}$ is observed when the concentration is increased to a certain value. For example, the $\lambda_{\mathrm{em}}^{\max}$ of the imine **2** solution is at 397 nm when the concentration is lower than 5 vol.%; however, it is increased to 457 nm when the concentration is more than 10 vol.%. While, for the imines **4** and **6**, their $\lambda_{\mathrm{em}}^{\max}$ slightly red-shifts or even remains constant with an increase in concentration. The appearance of two main $\lambda_{\mathrm{em}}^{\max}$ at different concentrations suggests the presence of two or more chromophores and the mutation of the chromophores with a change in concentration. This is further evidence of the aggregation of imine molecules and the formation of TSC among them. It is also necessary to note that, at low concentrations, the PL intensity of the solutions of imines **2** and **5** firstly increases quickly and then remains constant or even decreases with increasing concentration; however, when the concentration is further increased, the $\lambda_{\mathrm{em}}^{\max}$ red-shifts to a higher value and the PL intensity increases with increasing concentration. This abnormal phenomenon can also be explained with the transformation of chromophores. At low concentrations, with an increase in concentration, the number of chromophores increased, and, hence, the PL intensity increases. When the concentration is further increased, the chromophores gradually transform into another type, and hence the emission from the original chromophores decreases and that from the second type increases. The decreased PL intensity at high concentrations is possibly due to the self-screening effect of the solutions (Figure 3a).

Table 1 summarizes the λ_em_ of the imines in diluted, concentrated solutions and pure liquids as well as the quantum yields in solutions (5 vol.%). The $\lambda_{\mathrm{em}}^{\max}$ of all imines red-shifts with increasing concentration, and it is the highest in the pure liquids. The quantum yields of the imine solutions decrease with a decrease in TBC and increase gradually with a decrease in steric hindrance.

All imines also show excitation-dependent emission (EDE). Generally, with the increase in λ_ex_, the PL intensity firstly increases and then decreases, and the $\lambda_{\mathrm{em}}^{\max}$ gradually red-shifts (Figure 4). For the imines **2**–**5**, with the increase in λ_ex_, their $\lambda_{\mathrm{em}}^{\max}$ slightly red-shift first and then red-shift more significantly (Figure 4a–d). However, for imine **6**, with the increase in λ_ex_, a very slight red-shift in $\lambda_{\mathrm{em}}^{\max}$ is observed (Figure 4e).

### 2.3. PL Mechanism

The chemical structure and spatial arrangement affect the aggregate structures and the interactions among the chromophores of aldimines. Aldimines can adopt *cis*- and *trans*-configurations. The electronic structures and energy levels of the imines **1**–**6** in a single molecule state in both configurations were calculated (Figure 5, and Appendix A). The aromatic imine **1** has the lowest HOMO-LUMO energy gap of 4.29 eV and 4.34 eV in the *cis*- and *trans*-configurations, respectively, while the HOMO-LUMO energy gaps are significantly increased to 5.12 (5.18) eV and 5.40 (5.29) eV for the imines **2** and **3**, respectively. On the other hand, for the aliphatic imines **4**–**6**, their HOMO-LUMO energy gaps are obviously higher than those of the aromatic amines (≥1 eV), and there is a slight increase in the HOMO-LUMO energy gap as the substituent changes from cyclohexyl with a greater steric hindrance to a smaller *n*-butyl or isobutyl group. The reason for this is that cyclohexyl is a stronger electron donating group than butyl.

It is interesting to note that the change of the HOMO-LUMO energy gap is consistent with the change in the UV-visible absorption wavelength at a very low concentration (Figure 1a) and the chemical shift of the -N=CH group in ^1^H NMR spectra as well as the wavenumber of the C=N group in the FTIR spectra. With the increase in the HOMO-LUMO energy gap, the UV absorption peak blue-shifts, the chemical shift of -N=CH group decreases gradually, and the wavenumber of the C=N group in the FTIR spectra increases gradually.

The HOMO-LUMO energy gaps of the imines are higher than 4.29 eV (289 nm), which is higher than the excitation energies required for common photoluminescence. Therefore, the imines are not fluorescent emissive in a discrete state (dilute solutions). The formation of strong TSC becomes the dominant factor affecting the emissions.

Imine **1** has the lowest HOMO-LUMO energy gap, and TSC is also possible to form between the phenyl groups of different imine molecules in close contact. However, it is non-emissive in solutions or in a solid state. The single crystal data of aldimine **1** can be obtained from the literature [34]. Imine **1** crystal belongs to the monoclinic system of the P2_1_/c space group. As shown in Appendix A, in the crystal of imine **1**, the C=N group is coplanar with the phenyl group on the side connected to C, while the phenyl group on the side connected to N has a torsion angle of 55.2°. Therefore, no large planar π-conjugated structure can be formed. Moreover, the distance between the two adjacent phenyl that are parallel to each other is 7.921 Å, which is far beyond the distance that can form through-space π–π interaction. In addition, hydrogen bonding cannot be formed between the C=N and H-C= groups of adjacent molecules due to the steric hindrance of the phenyl groups. Therefore, there is nearly no TSC in imine **1** in the aggregates.

Different from imine **1**, the imines **2**–**6** exhibit intrinsic fluorescence and PL behaviors similar to other small molecular and polymeric NTLs. The difference in their PL behaviors should be attributed to the difference in their chemical structures (substituents on the C=N bond) and the intermolecular TSC in the aggregate states. For the aliphatic imines, their PL behaviors in aggregate states is completely dependent on the TSC of the C=N group between molecules, and the driving force for aggregation is intermolecular hydrogen bonding (C=N----H-C).

We calculated the energies of the optimized conformations of the *cis*- and *trans*-configurational isomers of the imines. As shown in Appendix A, the energies of the *cis-*isomers are always higher than those of the *trans-*isomers, suggesting that they tend to exist in *trans-*configurations in a single molecule state. ^1^HNMR characterizations also prove that the imines exist in *trans-*configurations. Unfortunately, the *trans-*configuration is not always the ideal configuration for which the imines to form a strong TSC in the aggregated state. We calculated the optimized conformations of the dimers of imines **2**–**4** in the *trans*- and *cis*-configurations (Figure 6). For the aromatic imines **2** and **3**, the intermolecular interactions in them are very similar for both *cis*- and *trans*-dimers. For imine **2**, the intermolecular interactions are mainly π–π interactions between adjacent phenyl groups (3.5~3.6 Å). Moreover, for imine **3**, the intermolecular interactions are composed of two parts: hydrogen bonding C=N----H-C (3.1 Å~3.3 Å) and H----π interaction (2.9~3.0 Å). For the dimer of imine **4** in the *trans*-configuration, the distance between the C=N groups of two adjacent molecules is 4.31 Å, and the length of the hydrogen bond is 3.216 Å. Therefore, no strong TSC and hydrogen bond can be formed. On the contrary, when the dimer is in the *cis*-configuration, the distance between the C=N groups is only 3.26 Å, and the length of the hydrogen bond is 2.785 Å. Similarly, the aliphatic imines **5** and **6** form aggregates through hydrogen bonds in the *cis-*configuration. These results suggest that aliphatic imines form aggregates in *cis*-configuration, though they adopt *trans*-configurations in a single molecule state.

The imines **2**–**6** show CDE and concentration-enhanced emission (CEE) behavior, suggesting that the aggregation of imine molecules plays an important role in their emission behaviors. We calculated the optimal conformations and HOMO-LUMO energy gaps of the dimer, trimer, and tetramer of the aliphatic imines **2**–**6** (Figure 7, and Appendix A), and the energy gaps are summarized in Table 2.

For the imines **2**–**6**, their HOMO-LUMO energy gaps gradually decrease with an increasing molecule number. The HOMO-LUMO energy gaps of the tetramers decrease to less than 5.00 eV for the aromatic imines and 6.00 eV for the aliphatic imines. Undoubtedly, the energy gaps should further decrease when the aggregates become larger. When the energy gaps are lowered to be less than the energies required for the excitation of the molecules with UV and visible lights, fluorescence emissions become possible.

The HOMO-LUMO energy gaps of the aromatic imines **2** and **3** are lower than that of imine **1** in a single molecule state; however, imines **2** and **3** are fluorescent emissive while imine **1** is non-emissive in aggregate states. The reason for this is that no strong TSC can be formed in the aggregates of imine **1** due to its *trans*-configuration and spatial conformation. On the contrary, the presence of an extra methylene between phenyl and C=N in imine **2** and the replacement of the phenyl group with the cyclohexyl group in imine **3** reduces the molecular rigidity; therefore, the molecules can adopt proper conformations to form strong intermolecular through-space π–π interactions between adjacent phenyl groups and/or between phenyl and C=N groups in the aggregates (Appendix A). The fluorescence emission of imine **3** is red-shifted in comparison to that of imine **2** in concentrated solutions and pure liquids. A possible reason for this is that a stronger TSC is formed in the concentrated solutions and pure liquid of imine **3**.

For the aliphatic imines **4**–**6**, the fluorescence emissions originate from the TSC formed by the C=N groups between molecules. The difference in the chemical structures of the aliphatic imines is the alkyl groups attached to the C=N group. The steric hindrance and electron-donating effect of the alkyl groups lead to a difference in the emission behaviors of the aliphatic imines. The cyclohexyl group has a bigger steric hindrance but a stronger electron-donating ability than the n-butyl and isobutyl groups, which leads to the red-shift of the UV absorption wavelength and the emission wavelength of the imines from 4 to 6 in diluted solutions with a decrease in the number of cyclohexyl groups. The quantum yield of the aliphatic imines increases with a decrease in the number of cyclohexyl groups, possibly due to the easier formation of the TSC for the imine with a smaller steric hindrance.

## 3. Materials and Methods

### 3.1. Materials

Aniline (99.0%), benzylamine (99.0%), cyclohexylamine (99.0%), *n*-butylamine (99.0%), benzaldehyde (99.0%), cyclohexanecarbaldehyde (97%), 2-methylbutyraldehyde (MBH, 98%), and deuterated chloroform-*d* (CDCl_3_, 99.8 atom% D, with 0.03 vol.% TMS) were purchased from InnoChem Technology Co., Ltd. (Beijing, China). Glycerol (99.0%), dichloromethane (CH_2_Cl_2_, 99.0%), and potassium hydroxide (KOH, ≥82.0%) were purchased from Beijing Tongguang Fine Chemical Co., Ltd. (Beijing, China). Magnesium sulfate (MgSO_4_, 99.0%) was purchased from Tianjin Kemiou Chemical Reagent Co., Ltd. (Tianjin, China).

### 3.2. Synthesis of Aldimines

The aldimines were synthesized by the reactions between primary amines with aldehydes (Figure 2) [35]. The mixture of amine (1 mmoL), aldehyde (1 mmoL), KOH (0.028 g, 0.5 mmoL), and glycerol (5 mL) was added to a round-bottom flask, and the reaction was stirred at a given temperature. The reaction mixture was washed three times with deionized water and extracted with 30 mL CH_2_Cl_2_. The organic phase was collected and then dried with anhydrous MgSO_4_. Finally, purified aldimine was obtained by vacuum distillation. According to the same reaction mechanism, different primary amines were reacted with different aldehydes to prepare six aldimines (Table 3). All aldimines are insoluble in water but soluble in common organic solvents.

### 3.3. Characterization

^1^H NMR spectra were recorded on a Bruker Avance III 400 MHz NMR spectrometer (Bruker BioSpin GmbH, Ettlingen, Germany), using CDCl_3_ as the solvent. FTIR spectra of the sample were recorded with a FTIR spectrometer (Affinity-1, Shimadzu, Kyoto, Japan), and the number of scans was 16 at a resolution of 1 cm^−1^. ESI+ mass spectrometry data were obtained by a high-resolution mass spectrometer (Triple TOFTM 5600+, AB SCIEX, Sun Valley, CA, USA). UV-vis spectra were recorded with a UV-vis spectrophotometer (UV-2450, Shimadzu, Kyoto, Japan).

### 3.4. Fluorescence Spectroscopy

Fluorescence emission and excitation spectra of the samples were measured with an FS5 fluorescence spectrometer and FS980 fluorescence spectrometer (Edinburgh Instruments, Livingston, UK). The excitation and the emission slit widths were 3 nm. The lifetimes were measured with an FS980 fluorescence spectrometer (Edinburgh Instruments, UK) and the luminescence quantum yields were measured with an absolute quantum yield tester (Quantaurus-QY, Hamamtsu, Japan). The photographs of the solutions under UV light (365 nm) were taken with a camera (D600, Canon, Tokyo, Japan) in a dark room. The exposure time was 1/4 s, and the ISO was 1600 unless otherwise stated.

### 3.5. Theoretical Calculation Methods

The conformation optimizations were performed with the density functional theory (DFT) based on Gaussian 09 software (Revision A.1; Gaussian, Inc.: Wallingford, UK) [36], and the lowest energy structure was assigned as the global minima. Frequency analyses were then carried out to confirm that the structures are minima. Finally, the energy calculations were performed with the density functional theory (DFT) with B3LYP functional and 6-31G(d,p) basic set with Grimme dispersion correction DFT-D3 [37,38,39,40]. The HOMO-LUMO electron cloud distributions were plotted with GaussView 6.0.

## 4. Conclusions

In this work, a series of small aromatic and aliphatic imines were synthesized, and their photophysical properties were studied and correlated to the intermolecular TBC and intermolecular TSC effects of the molecules in the aggregate state by theoretical calculations. Most of the imines exhibit bright photoluminescence in aggregate states, and their photoluminescence values show CDE, CEE, and EDE behaviors. The intermolecular TBC of aromatic imines is changed by replacing the phenyl group(s) with benzyl and cyclohexyl groups. The TBC extents of the aromatic imines are not large enough to induce fluorescence emission, and the formation of strong TSC becomes the dominant factor affecting the emissions. The photoluminescence of aliphatic imines is completely dependent on the intermolecular TSC of the C=N groups among molecules. The configurations and the spatial conformations of the imines play a decisive role in the formation of TSC. This study provides a preliminary understanding of the photoluminescence mechanism of NTLs and it will be helpful for the designing of novel photoluminescent materials.

## Data Availability

Not applicable.

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
