# Peer review of "Effects of Through-Bond and Through-Space Conjugations on the Photoluminescence of Small Aromatic and Aliphatic Aldimines"

_molecules, 2022, doi:10.3390/molecules27228046_

Round 1

Reviewer 1 Report

The work deals with an interesting problem which is the effects of through-bond and through-space conjugations on the photoluminescence of small aromatic and aliphatic aldimines. However, in my opinion, the Authors should consider some notable comments for future publication. Detailed comments are included below.

1. There is no information in the study which isomers of aldimines were obtained and which were tested: cis, trans or a mixture of both. No information about the trans or cis form dominating in dilute and more concentrated solutions, which would be confirmed, for example, by NMR. Maybe there are some literature data on this that should also be included.

2. Lines 16-19. The described two aromatic aldimines (2 and 3, cpd 1 is no-emitting) are not enough to make any general conclusions. Authors should investigate slightly more aromatic aldimines. Is the TBC of 3 really lower than the TBC of 2 as stated?

3. There is no description or any reference regarding imino-enol tautomerism for aromatic and aliphatic aldimines.

4. There is no calculated energy difference between the cis/trans isomers for aromatic aldimines (for the aliphatic the authors probably assumed it was similar to that for compound 4).

5. No attempt is made to explain why the theoretically most predisposed compound 1 does not absorb in dilute solution. Maybe this is the same reason it does not absorb as a solid. Explanation of non-absorption 1 in solid: "... one phenyl group is connected with the N atom of the C = N group by single bond, which can rotate freely in the discrete state. So, the excited state energy can be easily dissipated by intramolecular motion, leading to nearly non-emissive nature of imine 1 at room temperature ”can equally well be applied to the remaining compounds 2-6, which, however, absorb in dilute solutions.

6. There are no compound numbers in the figures 1-5, which would make reading easier. The LUMO HOMO structures are shown as cis and descriptions above them incorrectly as trans (Figure 5).

7. The absorbtion bands on the Figure S2(c) are not visible.

8. Lines 144-145. „In particular, the absorbance of imine 3 is the highest, partly due to the absorption of conjugated benzene ring and C=N structure. However, for the shoulder peaks appearing at around 400 nm, the absorbances of aldimines 3, 4 and 6 are higher than that of imine 2.” – Coupling of the aromatic ring with the CN group also takes place in the case of compounds 1 and 2, which absorbance is lower, therefore the argument presented is not appropriate. The descriptions are difficult to correlate with the spectra because the absorption bands are hardly visible (perhaps it would be better to clearly describe these bands).

9. Lines 144-147. What about compound 5? No uv-vis spectrum and no comment.

Reviewer 2 Report

I read the manuscript with interest. I think that this work is worth publishing in the journal; however, minor revisions are requested as listed below.

Figure 1b

Add compound number 4 under the compound structure.

Figure 2a1

Add compound numbers in the figure.

Figure 2

The “a1” should be “a”, “a2” should be “b”, “b” should be c, and “c” should be “d”.

Figure 3a

Add concentrations in the figure.

Figure 3b-3f

Add compound numbers in the figure.

Figure 6

The numbers of distance are too small.

Page 8, Line 277

The abbreviation “CEE” needs explanation.

Page 9, Line 305

The fluorescence emission of imine 3 is more red-shifted than…

All the compound numbers in the manuscript should be bold.

Author Response

Reply: Many thanks for your kind comments and suggestions on the improvement of our manuscript.

  1. All issues related to the figures in the manuscript and Supporting Information have been addressed.
  2. The full name of the abbreviation “CEE” (concentration-enhanced emission) has been added.
  3. Page 9, Line 305. The sentence has been revised as “The fluorescence emission of imine 3 is red-shifted with comparison to that of imine 2…”.

Round 2

Reviewer 1 Report

I would like to comment only on the main problem of this work and confirm my negative assessment. The presented research explain the increasing fluorescence under the influence of increasing concentration with the formation of aggregates in more concentrated solutions (>1%), while the emissive aggregates, as shown by calculations, are formed with cis-imines. The authors' explanation: "we believe that the imines are in trans form in dilute and more concentrated solutions” in the context presented in the manuscript may confuse.

As confirmed by the authors, in “dilute” solutions only trans form is present, but the question is: at what concentration were nmr spectra taken? If it was a typical concentration for 1H NMR i.e. 1-2% then the cis form should also be present in the solution.

The idea that increased concentration forces a cis configuration is interesting, but not supported by additional experimental data such as 1H NMR, NOESY, the authors only referred to theoretical calculations. If the NMR (or other) experiment confirms the presence of the cis form in more concentrated solutions (>1%), the presented calculations will explain why these solutions fluoresce, but if they do not confirm the existence of the cis form, another explanation should be proposed.